# Greenhouse Gas (GHG) Emissions from Honey Production: Two-Year Survey in Italian Beekeeping Farms

**DOI:** 10.3390/ani13040766

**Published:** 2023-02-20

**Authors:** Arianna Pignagnoli, Stefano Pignedoli, Emanuele Carpana, Cecilia Costa, Aldo Dal Prà

**Affiliations:** 1Centro Ricerche Produzioni Animali—CRPA Soc. Cons. p. A., Viale Timavo 43/2, 42121 Reggio Emilia, Italy; 2CREA Research Centre for Agriculture and Environment, Via di Corticella, 133, 40128 Bologna, Italy; 3Institute of BioEconomy-National Research Council (IBE-CNR), Via Giovanni Caproni 8, 50145 Florence, Italy

**Keywords:** GHG emissions, honey production, transport, supplemental feeding, climate index

## Abstract

**Simple Summary:**

A life cycle assessment (based on ISO 14040 and 14044) considering the climate change (CC) impact category on beekeeping was performed. To this aim, for two consecutive years, data from beekeeping farms were collected, including data on annual honey production, other hive products, geographical locations of the apiaries, the processing infrastructure, technologies used, and the fuel and energy consumption. The overall LCA result was estimated at 1.44 kg CO_2_e/kg honey, with transport and supplement feeding as main contributors to greenhouse gas (GHG) emissions. Migratory beekeeping systems were found to be more impactful than nonmigratory ones. Results of a climate index indicated that the scarcity of rainfall seems to negatively affect the honey yield, as well as increase the provision of supplemental feeding and the amount of GHG emissions. Despite the study limitations, the results obtained provide interesting insight to improve the sustainability of beekeeping practices in light of the EU Green Deal and Farm to Fork strategies.

**Abstract:**

The objective of this study was to quantify the climate change (CC) impact of the honey supply chain in different beekeeping systems and farms, over two consecutive years. The CC impact category is quantified as kg CO_2_ equivalent and it evaluates the GHG emissions, mainly CO_2_, N_2_O, and CH_4_. The results ranged from 0.44 to 3.18 (*p* = 0.039) kg CO_2_e/kg honey with higher values in 2021 than 2020. The main contributors to climate change of the honey supply chain are represented by transport and supplemental feeding inputs. The beekeeping system (migratory or stationary) influenced CC: the contribution to CC for stationary farms was estimated at 0.58 kg CO_2_e/kg honey and 2.48 for migratory ones (*p* < 0.001). Given the close connection between honey yield and LCA results due to the unit of measurement of impact, i.e., kg of honey produced, an index was developed (wildflower honey climate index) as a simple benchmark tool for prediction of honey yield in the survey context. Using the data from the present study, we found that the index is positively related to honey yield (r = 0.504; *p* < 0.05) but negatively related to supplemental feeding (r = −0.918; *p* < 0.01) and overall carbon footprint (r = −0.657; *p* < 0.05). Further studies are needed to better explain the effects of weather on honey production, as well as environmental impact.

## 1. Introduction

Bees play a key role for human and environmental existence, by ensuring the presence of crops and wild plants through pollination [1,2,3]. Scientists have estimated that the value of insect pollination ranges from 150 to 300 billion/year [4,5,6] with honeybees responsible for 80% of crop pollination [7]. Furthermore, beekeeping provides market products with health-promoting properties (honey, nectar, beeswax, etc.) [8], and it generates an extra source of income for local communities [9,10].

Currently, a total of 92.3 million hives have been recorded around the world, including 17.6 million in Europe and 1.5 million in Italy. Honey production amounts to around 1.8 million tons in the world, with 283 thousand tons in Europe and 23 thousand tons in Italy [11]. Honey yield in Italy has been declining significantly over recent years, with a minimum of 8.8 to a maximum of 14.7 kg per hive in 2021 [12]; however, data on honey production show variation. The yield of honey depends on a succession of flowering plants to provide a regular supply of pollen and nectar over the season. Previous research has shown that bee species are affected by weather conditions in various ways. According to Holmes [13], climatic conditions play an important role in explaining production variations, but the climate effect on honey yield is complex and not well documented. Other authors in different geographical contexts, with data collected in the field [14,15] or combining weather and satellite data and remote monitoring of honeybee colonies [16], have recently developed predictive models for honey harvests. A recent study investigated the connection between foraging activity of honeybees and local weather conditions [14].

A better understanding of the relationship between bees and the weather, along with the development of specific predictive indices (i.e., honey yield, bee health status), could potentially help beekeeping farmers manage bees with respect to the flowering season and local climate.

Several studies identified the current climate change as the main cause of bee decline, due to shifts in the flowering period of nectariferous plants that are a major food source for bees [17]. The effects of climate change on honey production differ depending on local weather conditions [18,19,20,21,22].

Even so, bees today are highly threatened by several factors including land-use change (LUC) [23], pesticide use [24], and the spread of parasitic infections [25].

Beekeeping production can also be affected by migratory or stationary hive management [26,27]. Migratory and stationary practices typically differ in honey yield, distances between apiaries, food administration, and number of treatments. Therefore, the general process of honey production and the required input are very different in the two types of beekeeping farms [28,29].

According to local climatic conditions and beekeeping practices, life cycle assessment (LCA) studies report variable results. Kendal et al. [28] evaluated the impact of honey production in the US from 0.67 to 0.92 kg CO_2_ equivalent/kg of processed honey in South America. Mujica et al. [29] estimated a carbon footprint of 2.5 ± 0.17 kg CO_2_e/kg honey for production in Argentina, whereas Vásquez-Ibarra et al. [30] reported it at 0.40 in Chile. For Italian beekeeping, Arzoumanidis et al. [7] assessed that the production of 1 kg of orange-blossom honey causes 1.01 kg CO_2_ equivalent, having considered that 60% of the total environmental impacts are allocated to honey production, while 40% are allocated to pollination services. Some studies focused on the effect of climate change on the honey chain or the biodiversity impacts [31,32,33,34,35,36,37,38]; however, there are still few life cycle assessment (LCA) analyses on honey production. To our knowledge, no analysis has compared the environmental impacts of different beekeeping systems taking into account weather data. 

This article was based on previous research [39]; climate data and LCA analysis on honey production in 2020 and 2021 were analyzed. 

The main purpose of this study was to quantify the greenhouse gas (GHG) emissions of honey production of different beekeeping systems in two survey years. In addition, the project aimed to estimate if any correlation exists between climate pattern and the estimated environmental impact. LCA analysis represents a useful tool for improving the environmental sustainability of the beekeeping supply chain, by identifying the main impact phases, and it can result in the implementation of mitigation strategies under specific climate conditions.

## 2. Materials and Methods

The methodology used in the present research is the life cycle assessment, following ISO 14040 and 14044 international standards [40,41]. 

### 2.1. The Beekeeping Farms

The test sample consisted of six beekeeping farms situated in central–northern Italy in the Provinces of Bologna, Florence, and Reggio Emilia. These farms were selected to achieve the study scope because they differed in both climate conditions and beekeeping systems. Their main characteristics are reported in Table 1.

According to a previous report [12], Italian honey production is highly influenced by the commercial scale and beekeeping management system. Thus, we considered in the analysis different types of beekeeping systems to include all types of Italian honey production.

The amount of supplemental feeding provided differed in the two analysis years, as well as in the farms according to the beekeeping management. An additional objective of this study was to assess if there was any correlation between the use of input, such as supplemental feeding and medication, and climate patterns. 

The beekeeping farms were also divided into different climate areas, as observed by Pinna [42] according to the Koppen classification. Farm 4, Farm 5, and Farm 6, which are located in the Emilia plain areas, are influenced by a continental temperate climate because the average temperature of the hottest month exceeds 22 °C. Farm 2 located in the Tuscan–Emilian Apennines is affected by an oceanic climate (Cfb), whereas Farm 1, which is situated in the nearby hilly area, is more influenced by a temperate climate (Cfa). Farm 3, located in Tuscany, is influenced by oceanic, temperate, and Mediterranean climates, with the latter (Csa) being predominant due to the farm’s proximity to the sea. 

These farms produce different honey types according to the typical vegetation resulting from the different pedoclimatic conditions in which they are situated. For instance, Farm 4, Farm 5, and Farm 6, which are located in the lowland livestock areas of the Emilia-Romagna region, produce, among others, alfalfa honey thanks to the widespread cultivation of alfalfa crop related to the Parmigiano Reggiano cheese production; Farm 1 and Farm 2, which are situated in hilly and mountainous areas, specialize in chestnut honey production; in Farm 3, where the seasonal and daily temperature range is lower than in the other farms, thanks to the Mediterranean climate influence, black locust honey is richly harvested. In addition to the described monofloral honeys, all six farms have multifloral honey production.

As regards beekeeping management practices, Farm 1, Farm 2, and Farm 3 represent migratory systems (Ms) due to the transportation of variable portions of their hives to different apiaries. During the two study years, Farm 1 transported 70% of its beehives, whereas Farms 2 and 3 transported 3% and 40% respectively.

Farm 4, Farm 5, and Farm 6 are stationary beekeeping systems (NMs); consequently, they did not transport any of their beehives.

### 2.2. Life Cycle Assessment (LCA)

Life cycle assessment (LCA) is the methodology identified by the European Commission for measuring and evaluating the environmental impacts connected to the life cycle of a product, such as global warming, ozone depletion, smog creation, eutrophication, acidification, toxicity, and resource depletion [35]. Global warming is the main impact category used, representing the climate change impact caused by human activities [36].

In the last few decades, LCA analysis has been applied in many areas, including the agri-food industry. However, few studies have focused on beekeeping supply chains [37,38], and their results were closely linked to honey yield.

The methodology applied in the present study was based on Pignagnoli et al. [39], with some important advances concerning the database used, data collection, and calculations.

The analysis considered two productive seasons in order to increase the robustness of the results and provide more information about beekeeping production, taking into account some climate parameters. The 2020 original case study was performed ex novo, given that a new hypothesis was taken into consideration.

#### 2.2.1. Goal and Scope Definition

The main purpose of the present life cycle assessment (LCA) was to quantify the greenhouse gas (GHG) emissions of honey production of six beekeeping farms in two survey years.

#### 2.2.2. System Boundaries and ‘Functional Unit’

The supply chain under study was the honey production of six beekeeping farms in two different years. The flowchart used to describe the life cycle of honey production is illustrated in Figure 1.

An attributional approach, which considered *from cradle to farm gate* system boundaries, was adopted. The system boundary included all inputs necessary to produce 1 kg of honey, starting from the hive management stage to the honey extraction (honey was harvested as liquid honey extracted from the combs by centrifugal force). The impacts generated from transportation and processing after the extraction were not considered.

Similarly to other studies (e.g., [7,28,29]), the functional unit (FU) selected was 1 kg of raw honey without packaging. This FU was selected since it reflects the typical function of the system at the product unit level under study, which is based on ISO 14040.

#### 2.2.3. Life Cycle Inventory (LCI)

Primary LCI data related to the honey supply chain were collected firstly from questionnaires submitted to beekeepers and then checked directly through personal interviews. According to different types of apiary management, the hive management stage was divided into spring–summer (honey production phase) and winter (nonproductive phase) periods. This stage included some inputs that were the same for each farm and season, i.e., wood and paint for hive structure, as well as other inputs, which depended on the seasonality, climate patterns, and the farm management, such as supplemental feeding, transport for migratory beekeeping, and electricity to maintain the frames during the nonproductive phase.

Supplemental feeding is necessary especially in the winter or dry season, when natural bee foods, such as nectar and pollen, are not available, in order to provide energy and proteins for bees [25]. The amount of supplemental feed differed in the considered farms and in the 2 years analyzed, as shown in Table 1; this input was quantified using specific ingredients reported in the product label.

Information about medication products used to avoid, reduce, or mitigate the impact of pests or disease was also required.

For transport of migratory hive management, both beehive seasonal placement and trips for honey harvests and regular apiary inspections were evaluated; we did not consider the transport of stationary hives, which was almost zero.

Honey harvest trips performed by migratory beekeepers to enable the honey production process included the placement and collection of supers, the number and frequency of which depended on the nectar availability, which varied according to the year, season, and surrounding flora [27]. The specific information on the transport routes, such as dates, travel distances, fuel consumption, and vehicle type, was precisely assessed in the beekeepers’ survey. In order to obtain the exact quantification of mileage, beekeepers marked after each trip the kilometers traveled, and they reported it in the questionnaires. This protocol allowed us to quantify the transport input as accurately as possible.

In the subsequent analysis step, all inputs of the honey extraction process were included. The honey extraction technology included the following main unit operations: centrifugation, decanting, filtration, uncapping, and wax recovery for further melting. Some of the main inputs of this phase were electricity to operate machinery and materials which made up honey processing equipment. The electricity consumption data were obtained directly from the meters located in each farm’s laboratory. The LCI data inventory is reported in the Appendix A.

Secondary data and background agricultural processes were taken from the Agribalyse v.1.3 database (https://nexus.openlca.org/database/Agribalyse, accessed on 1 September 2022). Instead, Ecoinvent v.2.2 (https://ecoinvent.org/the-ecoinvent-database/data-releases/ecoinvent-version-2/, accessed on 10 December 2022) was applied for those processes not specific to the agri-food production, such as low-voltage electricity consumption, obtained by the Italian national grid. The methodology principles of Agribalyse database aim to provide a homogeneous and consensual LCI European database to support environmental labeling policies and to help the agricultural sector to improve its practice, following the key international guidelines (ISO, LEAP, and PEF) [43,44]. 

#### 2.2.4. Allocation

The allocation procedure can be applied on physical or economic relations [40], while energy content can also be considered [30].

As suggested by Tillman [45], we carried out a mass allocation taking into account the physical relationship between honey and beekeeping coproducts (beeswax, honeycombs, propolis, pollen, and royal jelly).

Table 2 reports the types of coproducts and allocation values of the six farms in the two analysis years, estimated from the weight ratio of honey on total beekeeping products (honey and coproducts). ‘Honey allocation’ values represent the amount of GHG emissions allocated to honey production, based on the mass of honey compared to the total production. Beeswax was produced by each farm in the two analysis years, usually reused as wax foundation sheets; hence, in this case, it was not considered as a coproduct except for Farms 3 and 6, where the beeswax was used in the field of cosmetics and homoeopathic products. Royal jelly and propolis were produced by Farm 1 for both analysis years and sold in the local store.

#### 2.2.5. Life Cycle Impact Assessment (LCIA)

The life cycle impact assessment (LCIA) aimed to evaluate the environmental impacts based on the life cycle inventory within the framework of the goal and scope of the study [29]. According to the objective of the assessment, among the different impact categories, we assessed only the contribution to climate change (CC), also called global warming potential (GWP), expressed as kg of CO_2_ equivalent, in which the GHG emissions (CO_2_, CH_4_^+^, and N_2_O) were converted. The LCIA method used for this purpose was the IPCC 2013 method [46], as suggested by the ILCD handbook guidance for climate change impact category [47]. The quantity and permanence of the potential greenhouse gas in the atmosphere were quantified and then compared to CO_2_, which has a GWP conventional value of 1. The greenhouse gas quantities are normally expressed in kg CO_2_ equivalent (kg CO_2_e), through a standardization operation based on global warming potentials. For example, the characterization factors of N_2_O and CH_4_^+^ are 265 and 30.5, respectively [46].

The quantification of GHG emissions was carried out on the openLCA v.1.7.0 software (https://www.openlca.org/, accessed on 10 December 2022), using mainly the Agribalyse database with the support of Ecoinvent v.2.2.

### 2.3. Statistical Analysis

Prior to analysis, the data population normality was verified using the Kolmogorov–Smirnov test with SPSS software (IBM Corp. Released 2016. IBM SPSS Statistics for Windows, Version 24.0. Armonk, NY, USA: IBM Corp.). The data were submitted to statistical analysis (one-way ANOVA) using SAS/stat package version 8.0 (SAS Inst. Inc., Cary, NC, USA), considering the effect of farm, year of survey, and beekeeping systems. A significance threshold of *p* = 0.05 was considered. The recorded data (meteorological data, yield, and principal LCA inputs) were analyzed using backward stepwise regression and then linear correlation analysis to predict the wildflower honey climate index (WHCI). Next, we applied linear correlation analysis to identify the relationship between the developed index based on precipitation data and environmental results (CC and main inputs), as well as yield (Table 3). Electricity and supplemental feeding were considered in the statistical analysis, whereas transport input was not included because it only appeared in the migratory beekeeping systems.

### 2.4. Development of a Prediction Index Based on Meteorological Data and Correlation between Environmental Impact and Honey Production

The data included the average annual wildflower honey yield and meteorological data for years 2020 and 2021. For the development of the climate index, we selected wildflower honey because it was produced by all beekeeping farms (14 apiary sites); for this honey type, the variable number of harvests per year was recorded. These data were used to generate, for each year, the mean annual wildflower honey yield per hive, the mean first harvest honey yield per hive, and the proportion of the annual yield taken at the first harvest.

The colonies were maintained at different sites in the Emilia-Romagna, Liguria, and Tuscany regions in Italy, and, at each of the apiary sites, the nearest weather stations were identified (4.7 km distance on average between station and apiary). Mean, minimum, and maximum monthly temperatures (°C), monthly precipitation sums (mm), and days with rain (number of days with at least 0.1 mm of rain) were obtained from the regional meteorological agencies [48,49,50], (Appendix A). The data were analyzed using a backward stepwise regression combining weather variables with honey yields. We also correlated wildflower honey yield with carbon footprint and with the wildflower honey climate index that we developed. Considering data from meteorological stations, beekeeping monitoring devices, and other models [13,18], we attempted to develop a climate index in order to investigate the annual yield differences recorded for the beekeeping farms in different sites. This index can also be used to predict honey production in similar agroclimatic contexts.

## 3. Results

### 3.1. Life Cycle Assessment Results

According to the LCA methodology, total inputs and outputs resulting from beekeeping systems were integrated into the life cycle inventory to quantify GHG emissions for 1 kg of processed honey in 2020 and 2021.

The contribution to climate change (CC) of six beekeeping farms representing specific beekeeping systems is reported in Table 2; Farm 1 (Ms) ranged from 2.18 kg CO_2_e/kg honey in 2020 to 4.19 kg CO_2_e/kg honey in 2021, Farm 2 (Ms) ranged from 1.79 in 2020 to 2.90 in 2021, Farm 3 (Ms) ranged from 1.57 in 2020 to 2.26 in 2021, Farm 4 (NMs) ranged from 0.42 in 2020 to 0.96 in 2021, Farm 5 (NMs) ranged from 0.47 in 2020 to 0.77 in 2021, and Farm 6 (NMs) from 0.37 in 2020 to 0.51 in 2021. Electricity, transport, and supplemental feeding were the most impactful inputs, globally accounting for 92% in 2020 and 96% in 2021 of total GHG emissions. The beekeeping equipment, water, wooden hive structure, medication, and other minor inputs represented less than 10% of the total environmental impact; for this reason, they are not reported in Table 3.

Examining the influence of the farm, beekeeping system, and analysis year on the carbon footprint results, some trends could be observed, as partially confirmed by statistical analysis.

From 2020 to 2021 the CC result increased by 40% ± 10%, whereas honey yield decreased by various percentages. Different values, depending on the beekeeping system, were found, with an overall variation between a minimum of 17% and a maximum of 77%, in terms of kg honey produced by one hive.

The contribution of supplemental feeding and transport also varied from 2020 to 2021; according to the beekeeper surveys, the amount of supplemental feeding increased by 76% on average, ranging from a 33% increment for Farm 6 to 20% for Farm 4. Instead, the transport input accounted for less than 12% total migratory (Ms) emissions in 2021 compared to 2020. Farm 2 was a migratory farm, which mostly reduced its transport emissions, due to the migratory production of only black locust honey in 2021 unlike the previous year.

The different environmental impacts of supplemental feeding and transport were due to the shifting of input management in the two analysis years at the farm and beekeeping system levels. In fact, as found in the beekeeper surveys, from 2020 to 2021, the beekeeping farms maintained or increased the amount of supplemental feeding, whereas all migratory farms reduced total kilometers for hive management.

Examining the differences between beekeeping systems, the honey yield of migratory farms (Ms) was lower than that of nonmigratory farms (NMs) in both survey years; consequently, migratory farms (Ms) produced more than NMs. The correlation between yield and LCA results was established since the impact was estimated on the basis of the quantity of honey produced; consequently, honey yield was the key factor in the results.

In 2020, the mean environmental impact for CC category was 1.85 kg CO_2_e/kg honey for Ms and 0.42 for NMs; in the following year, it was 3.12 kg CO_2_e/kg honey for Msand 0.75 for NMs. As regards honey yield, the yield of Ms ranged on average from 9.33 kg honey/hive in 2020 to 3.53 in 2021 compared to NMs, for which it ranged from 15.56 kg honey/hive to 7.35 in the two analysis years, respectively.

The influence of the farm and the beekeeping system factors on the results was confirmed by the significant *p*-values, i.e., 0.039 and <0.001, respectively.

The major environmental impact of migratory farms was mainly due to transportation, as well as the lower yield. Transport for hive management generated on average 0.89 kg CO_2_e/kg honey (47% of total Ms CC) in 2020 and 1.14 kg CO_2_e/kg honey (35% of total Ms CC) in 2021.

Supplemental feeding contribution also differed in the Ms and NMs; it was responsible for 0.36 kg CO_2_e/kg honey on average for Farm 1 and Farm 2 in the first year and 1.65 in the second year. Supplemental feeding input caused much lower emissions in the NMs, with a mean value of 0.02 kg CO_2_e/kg honey in 2020 and 0.22 in 2021, according to the amount of supplemental food provided.

Strictly in terms of feeding input, migratory Farm 3 was characterized by similar beekeeping management to NMs, accounting for 0.03 kg CO_2_e/kg honey in 2020 and 0.21 in 2021.

The electricity impact differed between beekeeping farms and in the considered time period.

For the first survey year, electricity represented the main impactful input for NMs, producing 0.34 kg CO2e/kg honey i.e., 82% of NMs CC, whereas it accounted for 35% of Ms CC. In the second year, the electricity contribution was reduced for all farms except Farm 6 and Farm 3, in which it always accounted for 50% and 87% of the total impacts, respectively.

In the analysis process, three phases of the beekeeping process were defined: hive management in winter, hive management in spring–summer, and honey extraction (Figure 2). The winter season represented the honey nonproductive phase, while the spring–summer season represented the productive phase.

In the migratory beekeeping farms, more than half of the total impacts were reallocated to hive management phases (summer and winter), while, in the nonmigratory beekeeping farms, most of the impacts were allocated to the honey extraction phase.

The impact of different production phases was a consequence of individual inputs, which varied depending on analysis year, beekeeping system, and farm, as previously explained. In the first analysis year, hive management in the summer was the main impactful phase in migratory beekeeping systems (Ms), providing 55% of total GHG emissions in Farm 1, 48% in Farm 2, and 43% in Farm 3, caused by the high number of trips for the collection of honey supers and frequent inspections of apiaries needed in the productive season. In the second year, the impact of summer hive management on total CC was reduced despite this phase producing more or equal kilograms of CO_2_ equivalent in 2021 compared to 2020 (see Appendix A). The outlined trend was caused both by the reduction in transport from 1 year to the next and by the increase in inputs for the most part needed in the nonproductive phase. Consequently, the nonproductive phase with supplemental feeding as the first input increased from 27% in 2020 to 53% in 2021 for Farm 1 and from 32% to 42% for Farm 2. The nonproductive phase emitted 0.45 kg CO_2_e/kg honey (24% ± 2% of total CC) in Farm 3, which provided the same low level of supplemental feeding in 2020 and 2021.

Electricity provided the main impact in the honey extraction phase due to the energy consumption required by the dedicated equipment; electricity was also necessary to preserve the farms in the winter period, but its impact was negligible.In the stationary beekeeping systems (NMs), the honey extraction was the most impactful phase (60% of total CC), whereas, in the migratory beekeeping systems (Ms), it was the least impactful one (20% of total CC).

The honey extraction contribution to Farms 4 and 5 CC decreased in the second year due to the greater accounting of the productive phase for the first farm and the nonproductive phase for the second one. On the other hand, in the case of Farm 6, this phase emitted more GHG emissions per kg of honey produced, 0.24 kg CO_2_/kg of honey in 2020 vs. 0.41 in 2021, i.e., 64% of CF in 2020 and 79% in 2021.

### 3.2. Development of a Prediction Index and Correlation between Environmental Impact

At first, data from all weather stations of hive sites were used (more details are given in Appendix A), yielding interesting trends. However, the equation was not significant (*p* = 0.165) for the average honey produced by the six farms in the 2 years of survey; the equation was also not significant for other types. Consequently, the index was developed on the basis of the climate and productive data of wildflower honey, which was the only honey type produced by all beekeeping farms surveyed in this study.

Temperature and precipitation data were analyzed initially using Pearson (linear) correlation; all parameters related to the increase in rainfall showed a positive correlation with the production of wildflower honey. Temperature data were found to be nonsignificant and were removed from the index. The cumulative rainfall, days with rain (minimum accumulation 0.1 mm), days with bee activity, and days with precipitation from 1 May to 30 August were instead included in the following index:* Wildflower honey climate index = −[a + b + c + (d × 10)/100],(1)
where a is the cumulative rainfall (millimeters), b denotes the number of days with rain (minimum accumulation 0.1 mm), c denotes the number of days with bee activity (counting days with average temperature > 10 °C), and d denotes the number of days with precipitation from 1 May to 30 August (number of days × 10).

The wildflower honey climate index was validated on six beekeeping farms in the area, and the honey yield declared by the beekeepers consistently correlated with the regression line developed. No reliability could be attributed to the index in the case of bee diseases, i.e., *Varroa destructor*, or other events with a significant impact on the hive. However, it is clear from this index that many aspects of weather influence wildflower honey production. The index developed in the six beekeeping farms and in the 2 years of survey, applied to our data, recorded the highest value of −11.14; in this case, the expected production would be ≥11.76 kg of wildflower honey per hive. The lowest recorded index value was −23.27; in this case, the estimated yield would be ≤2.20 kg of wildflower honey per hive. This index can be used as a benchmark yield compared to those recorded in our study.

We found that the developed index was positively correlated (r = 0.504; *p* < 0.05) with the yield, expressed as kg of wildflower honey per hive of the six beekeeping farms for 14 apiary sites (see Figure 3).

On the other hand, we highlighted negative correlations between our wildflower honey climate index and the carbon footprint (r = −0.657; *p* < 0.05) and the supplemental feeding (r = −0.918; *p* < 0.01). Thus, as the index increased, the impacts of carbon footprint and supplemental feeding tended to decline. Through the correlations, it emerged clearly that there were two different clouds of data coming from stationary (higher yield, lower carbon footprint, and lower use of supplemental feeding) and migratory beekeeping farms; this aspect related to yield has been less defined.

These results, strictly related to the climatic area analyzed, show that precipitation as the basis of the climate index caused an increase in honey yield and a decrease in LCA results and supplemental feeding. The climate index described was developed considering that the present study focused on a beekeeping life cycle assessment; further investigations may better explain the data obtained, considering other climate parameters and including the morphological characteristics of bees.

## 4. Discussion

### 4.1. GHG Emissions from Honey Production

The greenhouse gas (GHG) emissions in our survey sample were estimated to be 1.13 kg CO_2_e/kg honey in 2020 and 1.93 kg CO_2_e/kg honey in 2021, with an average value of 1.44 kg CO_2_e/kg honey. These values represent the average of the results reported in Table 3. On this basis, some conclusions can be drawn; the results are concordant with those reported in other LCA studies with some exceptions due to the different analysis methodologies and beekeeping systems tested. For example, Moreira et al. [50], in a study conducted on honey production in Spain, reported an impact of 1.66 kg CO_2_e/kg of honey, except for two producers, for which impacts of 5.96 and 3.20 kg CO_2_e/kg of honey were estimated. Overall, the average result of the reported studies was 1.20 kg CO_2_e/kg honey without final packaging, which was also excluded from our system boundaries.

Arzoumanidis et al. [7] (1.01 kg CO_2_e/kg honey) and Kendal et al. [28] (0.72 kg CO_2_e/kg honey) included the assessment of pollination services and found lower LCA results for the CC impact category than our study, due to the application of different allocation methods (economic allocation instead of a mass allocation ) and different beekeeping practices considered. It follows that the allocation method may significantly affect the results.

Mujica et al. [29] estimated 2.5 ± 0.17 kg CO_2_e/kg honey for Argentinian beekeeping production; however, the production of Argentine honey differs from the Italian one. According to Argentinian beekeeping production, the bulk of honey supers are collected from each farm and then processed in centralized facilities, which have capacities ranging from a minimum of 400 to a maximum of 700 kg honey/h [26]; consequently, electricity was the most impactful input, with a higher CF than our assessment.

The GHG emissions quantified in this study varied according to the beekeeping system and analysis year. The migratory beekeeping system was significantly more impactful than the nonmigratory beekeeping system, accounting for 2.50 kg CO_2_e/kg honey compared to 0.58 kg CO_2_e/kg honey, respectively.

The different results can be explained mainly by the honey yield and transport input related to the Ms farms. Yield and beekeeping systems significantly affected the results, as found in the statistical analysis.

According to the beekeeping surveys, the migratory honey yield was half that of the nonmigratory one in 2020 as well as in 2021, with mean values of 6.45 kg of honey in the hive compared to the 11.45 kg produced by nonmigratory farms. The productivity of the honey of each apiary plays a fundamental role since it represents the functional unit on which environmental impact is quantified. Thus, it has a direct effect on the functional unit, as assessed by LCA studies [28,29,51].

Transport input represented the main contributor to climate change on average for 2020 and 2021; as a consequence, the migratory farms resulted more impactful, whereas the nonmigratory farms achieved significantly lower emissions due to their zero transportation needs. Kendal et al. also obtained the same results in their LCA research [28,32].

Supplemental feeding was one of the main impactful inputs, and its contribution changed in the two analysis years, increasing the environmental impact estimates by 24% for both Ms and NMs in 2021 due to the greater amount of feeding provided, as reported in Appendix A.

The relevance of transport and feeding in the LCA assessment performed on the climate change category was confirmed by other studies; Vásquez-Ibarra et al. [30] identified feeding as the primary impact in Chilean beekeeping systems, whereas Moreira [51] and Kendal [28] assessed transport as the most impactful activity in the beekeeping supply chain.

Electricity for honey extraction was the primary impact in NMs, but it globally accounted for less than 40% of the overall impact, in contrast to Mujica et al.’s results [29].

According to single inputs, the most impactful phase of honey production was the productive phase for migratory beekeeping systems, caused by hive control trips mainly occurring in the summer season; honey extraction for nonmigratory beekeeping systems was primarily impacted by electricity.

As suggested by Vásquez-Ibarra et al. [30], the differences with other LCA studies could be due to the different productive activities and types of inputs considered, with values often dependent on local conditions.

Considering the outcomes of these results, the sustainability of the honey supply chain could be improved through best management practices mainly focused on transport and supplemental feeding.

One possible method to reduce GHG emissions is to use more efficient transport modes and minimize transport distance, planning the number and date for visits in order to increase the number of activities in hives per visit [30]. On the other hand, we suggest using, in supplemental feeding, ingredients which have a better environmental performance with the same nutritional properties. For example, the Agribalyse database reports for European consumption a lower impact for beet sugar than cane sugar.

Innovative technologies and modern technical facilities could support beekeepers, improving the efficiency of beekeeping and facilitating their work [52,53,54].

Currently, various systems based on temperature, sound, and video real-time monitoring have been developed and applied in the beekeeping industry, following the precision agriculture principles.

Precision beekeeping combines information technologies and beekeeping science and is defined as an apiary management strategy based on the monitoring of individual bee colonies to minimize resource competition and maximize the productivity of bees [51,52].

### 4.2. Climate Influence on Honey Yield and GHG Emissions

The causes of honey yield differences recorded in the two analysis years, i.e., 12.4 kg of honey per hive in 2020 compared to 5.4 in 2021, were further investigated through the development of a climate index. Considering the key role of honey yield in the life cycle assessment, the investigation of climate effect on productivity may impact the environment, as well as the beekeeping sector.

Scientific studies have identified different climate factors that influence honey yield, directly by impacting bees or indirectly by impacting flowering patterns [13,14,18,21,22].

The possible mismatch between the phenology of plants and honeybees, the introduction of new cultivars (i.e., high-oleic sunflower), and the loss (i.e., Sulla and Sainfoin) of annual crops bred in response to a changing climate are of concern to beekeepers [55,56,57]; the management of early cutting (flower button) of alfalfa also limits the pabulary resources of bees and, consequently, honey yield.

The first phenological sign of spring activity in a honeybee colony is the first cleansing flight that bees perform to avoid after the winter months, which is dependent on the late winter temperature, as suggested by Langowska et al. [21].

For this reason, in three beekeeping farms representative of the climate context of the survey, beekeeping monitoring devices were installed to obtain data about bee activity, such as the resumption of bee activity in spring through the change in weight of beehives. The definition of the parameter ‘days with bee activity (counting days with average temperature >10 °C)’, together with the other climate indices (cumulative rainfall and days with rainfall), allowed us to develop a prediction model of the production of wildflower honey. The climate index showed that precipitation had a positive effect on honey production, as well as on reducing GHG emissions and the supplemental feeding impact of the six beekeeping farms in the 14 sites surveyed in this study.

Climate change had various effects on honey production according to local weather conditions. The positive correlation between precipitation and honey production found in this research was confirmed by Gajardo-Rojas et al. [20], who assessed that drought conditions in the Mediterranean region of Chile have caused a decrease in honey yield since 2010. On the other hand, Solovev et al. [22] found that the absence of precipitation led to larger nectar gains and higher beekeeping productivity in the northwest Russian Federation, whereas they also assessed that precipitation is often necessary in hotter and dryer regions for nectar production.

The scarcity of rainfall made it necessary to provide more supplemental feeding in our surveyed sample, as well as in other Italian beekeeping contexts [11]. This necessarily caused an increase in GHG emissions, while also resulting in a reduced yield, as shown by the results of the LCA and statistical analyses.

Certainly, the proposed index can be improved by the inclusion of other meteorological data that affect the ethology of bees. One example is wind data; unlike the Holmes study [13], we did not consider this aspect due to the fragmented data that were freely accessible. Additionally, in the future, the morpho-productive characteristics of bee families, such as wintering resistance, queen prolificity, colony strength, disease resistance, and brood viability, can be considered in the model, as developed by Cebotari et al. [18].

Despite its limitations, to our knowledge, this is the first study to carry out an LCA analysis while also considering environmental data; the correlations between the climatic conditions and the main climate factors influencing the honey GHG emissions highlight the vulnerability of the bee industry, which is already exposed to many critical issues in Italy [58,59].

## 5. Conclusions

The environmental honey carbon footprint quantified in six beekeeping farms for two analysis years ranged from 0.44 to 3.18 kg CO_2_e/kg honey. Variations were mainly due to transport and supplemental feeding. A higher impact value was quantified for 2021 compared with 2020, which was also influenced by the lower honey production reported in the second year.

In our survey, the type of beekeeping system significantly affected the results. The migratory beekeeping system emitted more GHG emissions than the stationary one. The hive management phase, especially in the spring–summer season, was the most impactful phase of the beekeeping supply chain.

Thus, the environmental impacts of honey production could be reduced through management practices mainly focused on feeding and transport, using more efficient transport modes and less impactful ingredients in supplemental feeding.

The current results also highlighted an influence on the kg of honey produced. Many aspects of weather influence the wildflower honey yield; we found that the scarcity of rainfall was connected to a lower honey yield, as well as to a higher impact of supplemental feeding and a higher overall estimation of GHG emissions.

In conclusion, we suggest that the use of technology in beekeeping practices could improve the environmental beekeeping performance while also considering the climate factors. In this case, the implementation of a smart apiary management in beekeeping management can be a useful tool for efficient resource management, minimizing control trips and providing supplemental feeding at the necessary times and in the necessary amount.

## Figures and Tables

**Figure 1 animals-13-00766-f001:**
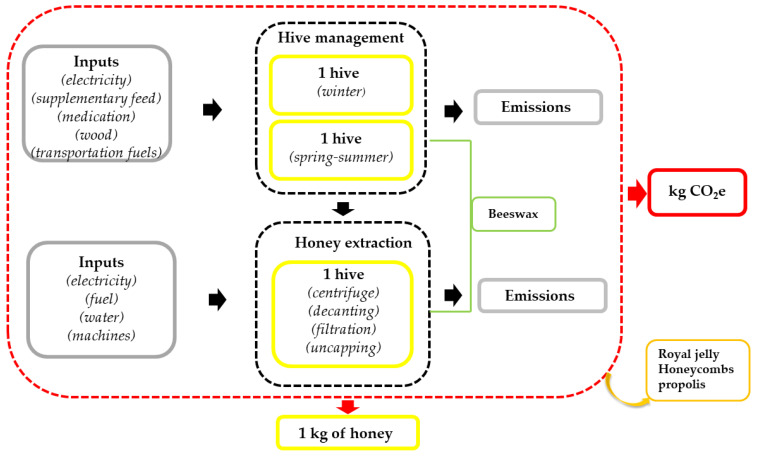
Flowchart of the boundary and main activity for the LCA beekeeping chain.

**Figure 2 animals-13-00766-f002:**
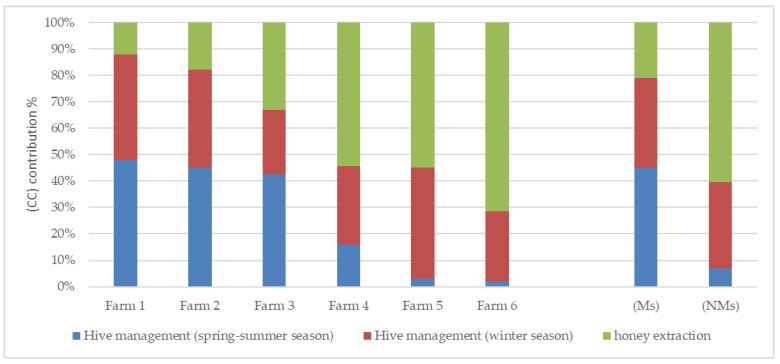
Contribution to climate change (CC) of different beekeeping phases for six farms analyzed in 2020 and 2021.

**Figure 3 animals-13-00766-f003:**
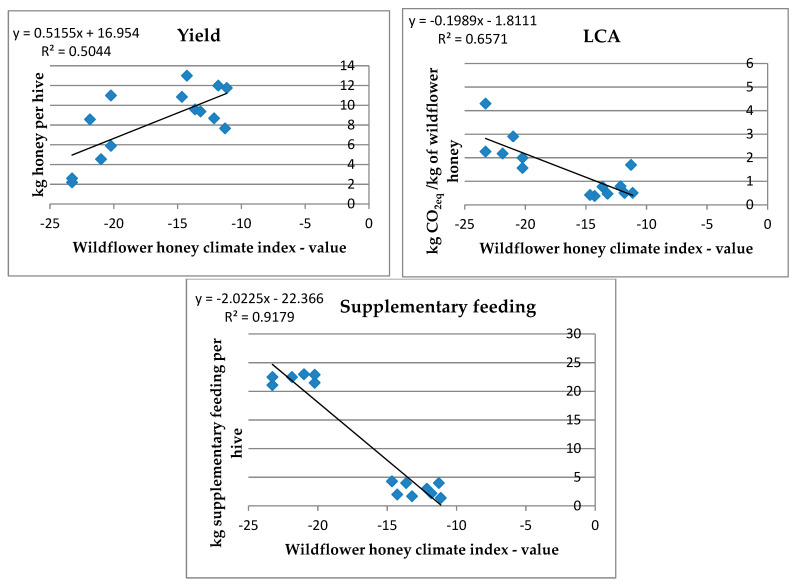
Statistical correlation between wildflower honey climate index and yield, LCA results concerning CC impact category, and supplemental feeding.

**Table 1 animals-13-00766-t001:** Beekeeping farms analyzed in the study case.

Beekeeping Farms	Climate Classification	Supplemental Feeding 2020	Supplemental Feeding 2021	Beehives Transported 2020	Beehives Transported 2021
Farm 1	Subcontinental temperate (Cfa)	High	High	80%	60%
Farm 2	Oceanic (Cfb)	Medium	High	16%	8%
Farm 3	Hot summer Mediterranean (Csa)	Low	Low	34%	47%
Farm 4	Subcontinental temperate (Cfa)	Low	Medium	0%	0%
Farm 5	Continental temperate (Cfa)	Low	Low	0%	0%
Farm 6	Continental temperate (Cfa)	Low	Low	0%	0%

**Table 2 animals-13-00766-t002:** Coproducts and allocation percentage on honey production.

Beekeeping Farms	Coproducts 2020	Honey Allocation 2020	Coproducts 2021	Honey Allocation 2021
Farm 1	Beeswax Honeycombs Propolis Royal jelly	93%	Beeswax Honeycombs Propolis Royal jelly	93%
Farm 2	Beeswax Honeycombs	99%	Beeswax Honeycomb	87%
Farm 3	Beeswax	99%	Beeswax	99%
Farm 4	Beeswax	100%	Beeswax	100%
Farm 5	Beeswax Honeycombs Royal jelly	88%	Beeswax	100%
Farm 6	Beeswax	97%	Beeswax	96%

**Table 3 animals-13-00766-t003:** Contributions to climate change (main impact and total) of honey production in the 2 years of survey.

Farm	System	Year	Electricity	Supplemental Feeding	Transport	Honey Yield	(CC) Contribution						
							Honey	Farm		System		Year	
			% of Total Impact	% of Total Impact	% of Total Impact	Kg of Honey/ Hive	Kg CO_2_ e/ kg of Honey	Mean ± s.e	*p*-Value	Mean ± s.e	*p*-Value	Mean ± s.e	*p*-Value
1	Ms	2020	24	16	56	11.45	2.18	3.18 ± 0.77	0.039	2.48 ± 0.36	<0.001	1.13 ± 0.31	0.265
		2021	11	38	48	2.47	4.19						
2		2020	31	22	43	9.81	1.79	2.34 ± 0.42					
		2021	18	59	20	2.58	2.90						
3		2020	49	2	43	6.73	1.57	1.91 ± 0.26					
		2021	50	9	37	5.60	2.26						
4	NM	2020	75	8	0	10.27	0.42	0.69 ± 0.20		0.58 ± 0.09		1.93 ± 0.59	
		2021	47	49	0	6.53	0.96						
5		2020	84	7	0	15.0	0.47	0.62 ± 0.11					
		2021	53	39	0	4.53	0.77						
6		2020	87	5	0	21.42	0.37	0.44 ± 0.05					
		2021	87	8	0	11.0	0.51						

## Data Availability

Not applicable.

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
