# Peer review of "Greenhouse Gas (GHG) Emissions from Honey Production: Two-Year Survey in Italian Beekeeping Farms"

_animals, 2023, doi:10.3390/ani13040766_

Round 1

Reviewer 1 Report

Overall, the authors present a very interesting case study. However, there certain issues that have to be improved. The main methodological issue is that the Functional unit is not correct. Moreover, the inventory table for the examined lifecycle is not provided. Also, the allocation ratios among the various process by-products should be provided.

Specifically:

Remove the first two paragraphs of the Introduction (Lines 44-53). Your research is on beekeeping, not on LCA. You can place those two paragraphs later on, in the beginning of section 2.2. Please renumber all references accordingly.

Lines 153-155: the definition of the functional unit is completely wrong. CO2eq. is an index for the environmental impact "climate change". The functional unit should be "1 kg of honey produced". Please correct it.

Lines 162-163: what percentage of the total honey producers, does your sample represent?

The life cycle inventory table is missing. You should include data for all six farms in this table. This is a major drawback of the manuscript.

How did you model the supplemental feeding  in terms of CO2eq.? Which is the electricity mix for Italy considered in your analysis?

Lines198-199 "...consequently mass allocation was applied by quantifying co-products ratio on total honey production." Please mention the numeric ratios of the allocation process.

Lines 251-253. please remove the following paragraph "This section may be divided by subheadings. It should provide a concise and precise description of the experimental results, their interpretation, as well as the experimental 252 conclusions that can be drawn. " I guess it is from the instructions for authors.

Minor issues:

Lines 362-363: please replace numbers below ten with words, not numbers e.g.: six farms, two years

Author Response

Author's Reply to the Review Report (Reviewer 1)

Overall, the authors present a very interesting case study. However, there certain issues that have to be improved. The main methodological issue is that the Functional unit is not correct. Moreover, the inventory table for the examined lifecycle is not provided. Also, the allocation ratios among the various process by-products should be provided.

Authors - Thank you for your suggestion, we have corrected the functional unit, we also included the allocation rations in the text with Table 2 and the Life Cycle Inventory in the Supplementary Tables 1,2,3.

Specifically:

Remove the first two paragraphs of the Introduction (Lines 44-53). Your research is on beekeeping, not on LCA. You can place those two paragraphs later on, in the beginning of section 2.2. Please renumber all references accordingly.

Authors - Suggestion accepted, thank you.

Lines 153-155: the definition of the functional unit is completely wrong. CO2eq. is an index for the environmental impact "climate change". The functional unit should be "1 kg of honey produced". Please correct it.

Authors - Sorry for the typing error, the sentence has been rewritten as suggested, thank you.

Lines 162-163: what percentage of the total honey producers, does your sample represent?

Authors - The farms analysed are located in different Italian regions and they are characterized by different climate conditions as we explained in the Section 2.1. From a national point of view, in Italy the apiaries are recorded 153339, for a total of 63408 beekeeping activities. The region with the greatest number of apiaries is Piedmont, with 23900 units, followed by Lombardy (17303) and Emilia Romagna (14598). (Anagrafe Nazionale Zootecnica, 2021).

The life cycle inventory table is missing. You should include data for all six farms in this table. This is a major drawback of the manuscript.

Authors - Suggestion accepted, we have added this table in the supplementary materials (Supplementary tables 1,2,3).

How did you model the supplemental feeding in terms of CO2eq.? Which is the electricity mix for Italy considered in your analysis?

Authors - To calculate the amount of supplemental feeding we have considered the specific formula reported by label of product. The Agribalyse flow used is ‘beet sugar production, all. default, U - CH’. For Electricity input we have used the Ecoinvent flow ‘Electricity, low voltage - IT’ as suggested by other authors, for example Arzamoudis et al., 2019.

Lines198-199 "...consequently mass allocation was applied by quantifying co-products ratio on total honey production." Please mention the numeric ratios of the allocation process.

Authors - Suggestion accepted, we have included the allocation values in Table 2.

Lines 251-253. please remove the following paragraph "This section may be divided by subheadings. It should provide a concise and precise description of the experimental results, their interpretation, as well as the experimental 252 conclusions that can be drawn. " I guess it is from the instructions for authors.

Authors - Sorry for the typing error, we have corrected the sentence.

Minor issues:

Lines 362-363: please replace numbers below ten with words, not numbers e.g.: six farms, two years

Authors - We found the typo and corrected it (line 404), thank you.

Reviewer 2 Report

Authors of this manuscript attempts to assess the GHG emissions of beekeeping, distinguishing between two different beekeeping management (i.e. migratory and non-migratory systems). Furthermore, they develop an index linking some precipitations parameters and bees’ activity (number of days with a temperature higher than 10°C). Such index is correlated with the then wildflower honey yields and with the GHG emissions found.

To my opinion, these are topics of relevance and general interest to the readers of this journal. However, the manuscript in the current form is not acceptable for publication.

I list hereunder the main general issues that weaken the work.

1)      The novelty of this work is limited. Indeed, several life cycle assessment (LCA) analysis and carbon footprint (CF) of honey production have been published in literature (as appear in the references’ list) and the correlation between weather parameters (such as precipitation and temperature) with honey production has been also investigated by several scholars.

Authors clearly state that the current work builds upon a previous work of them (Pignagnoli et al. 2021, cited by authors). Checking the paper of Pignagnoli et al. 2021 it appears that the analysis performed concerning the GHG emissions evaluation is almost the same, but including data from a second year (the ones of 2021). The results of 2020 of the present work are the same of the one published by Pignagnoli et al. 2021. Furthermore, the entire text of section 2.2.1 is almost identical of the one present in section 2.4 of Pignagnoli et al. 2021. I found too high similarity of several other sentences and paragraph between the two manuscripts.

The fact that a part of the results has been already published in another journal, as well as that entire sentences are almost the copy of the ones present in a published article, doesn’t represent the standards of publication ethics requested by this Journal.

2)      I found serious inaccuracies and imprecisions about the methodology description, in both the GHG emissions evaluations and in the index definition, which make me doubt about the accuracy and correctness of their implementation.

Concerning GHG emissions evaluation

Carbon footprint (CF) and life cycle assessment (LCA) are different concepts and different methodology are used to evaluate them. In the whole document these terms are instead used interchangeably, confusing the reader and making unclear what the aim of the study is. Indeed, first it is said that the aim of the study is to conduct a LCA (e.g. line 22, 142-143), but later (e.g line 24, 145-156) carbon footprint is mentioned. Or even “In order to assess the environmental impact of honey production, ..” (line 153) making the reader struggling to understand to which environmental impact authors are referring to.

A definition of CF is given by authors in line 148-149 (although not in a clear form), but it is not what they assessed, since they didn’t include only “carbon dioxide emission” (line 148), rather certain GHG emissions (such as CO2, CH4 and N2O), as stated in line 153-155.

In the case authors performed a CF, they are invited to consult and follow the recent standard ISO 14067: 2018, which has been developed specifically concerning the Carbon footprint of products.

In the case authors performed a LCA, providing the results only for the climate change (or global warming) impact category, they are invited to clearly state it, using the right wording and definitions.

In addition to this, at line 142-144 it is said that recent updates at the international level have been considered concerning the methodology used. To which updates are they referring to?

In LCA, as well as in CF, the functional unit is defined as “ performance of a product system for use as a reference unit. …. This functional unit defines what is being studied. All subsequent analyses are then relative to that functional unit, as all inputs and outputs in the LCI and consequently the LCIA profile are related to the functional unit.” (ISO 14044), that means “the functional unit that names and quantifies the qualitative and quantitative aspects of the function(s) along the questions “what”, “how much”, “how well”, and “for how long”. For a product this could be e.g. "Complete coverage of 1 m2 primed outdoor wall for 10 years at 99.9 % opacity").” (ILCD-Handbook 2010). Indeed, the FU used by authors is 1kg of honey in one year, and not kg CO2eq/kg honey, as wrongly reported by authors in line 154.

Authors probably confuse and mixed up the inventory phase with the impact assessment phase. They named section 2.2.2 as Life Cycle Inventory Assessment (LCIA), which doesn’t correspond to any LCA phase. The ISO norms, which authors declare to follow, indicate four mains fundamental LCA phases: goal and scope definition, inventory analysis (or life cycle inventory - LCI), impact assessment (or life cycle impact assessment – LCIA) analysis and interpretation. Section 2.2.2 should be correctly named (i.e. life cycle impact assessment) and only information concerning the impact assessment phase should be provided (e.g., the fact that the database Agribalyse was used and that royal jelly and propolis was produced only by farm 1 (lines 215-223), should positioned in a section where the inventory is presented and discussed).

Generally, LCA studies, as well as CF studies, report a quantitative list of input-flows (e.g. number of hives, quantity of wood used for each beehive, transport distances and frequency, fuel consumptions, electricity consumption, etc). None of this information is provided, making the analysis impossible to be replicated and verified. Also the whole beekeeping management systems are poorly described.

Only one farm produces royal jelly, representing a peculiar situation among the farms investigated. To produce royal jelly, beehives are managed in a completely different way compared to ones dedicated to honey production. Why authors decided to include them in the assessment? I would suggest excluding the beehives dedicated to royal jelly production from the analysis, since the core object of the study is honey, and not royal jelly or the whole bee-farm products.

Concerning allocation, since already in Pignagnoli et al. 2021 emerged that the honey yields significantly affect the results, authors could have estimated and presented the result using an additional FU, such as for example 1kg of beehives product in one year. In this way allocation would have been avoided, as suggested by the ISO and by several authors.

Concerning the index development

The “wildflower honey climate index” is mentioned along the manuscript without giving any introduction about it. Indeed, within the introduction, as well as along the whole text, no basic and useful information are given, information on possible similar indexes are not provided to the reader, and an explanation on the usefulness of this index is missing.

Several papers exist showing the influence of weather conditions (e.g. precipitation, temperature, relative humidity, wind speed, solar radiation) on honey production and on bees activities

It would be beneficial to develop further the index proposed in this study, including more weather-linked parameters and other fundamental aspects affecting honey production, such as the presence and severity of pathogens and disease, the availability and abundance of food sources (e.g. pollen, nectar, or honeydew produced by other insects), the ecological structure of the surrounding (e.g. fragmented landscape, natural areas, cultivated fields, urban centres, etc), possible polluting sources, etc. Some inspirations can be taken for example from additional recent literature:

Gounari, S., Proutsos, N., & Goras, G. (2022). How does weather impact on beehive productivity in a Mediterranean island?. Italian Journal of Agrometeorology 1: 65-81. https://doi.org/10.36253/ijam-1195;

Campbell, T.; Dixon, K.W.; Dods, K.; Fearns, P.; Handcock, R. (2020) Machine Learning Regression Model for Predicting Honey Harvests. Agriculture 10, 118. https://doi.org/10.3390/agriculture10040118;

and from Clarke, D., Robert, D. (2018) Predictive modelling of honey bee foraging activity using local weather conditions. Apidologie 49:386–396. https://doi.org/10.1007/s13592-018-0565-3.

Authors state that the index was developed to predict the production of honey (line 32). Did authors tested in some way the index they developed? Are the predicted results matching the measured results?

Authors state that the monitoring wages were placed under only three beehives (lines 244-245). Why only three and not maybe six beehives, one for each farm? Or even better, one for each site? How these three beehives have been selected? Where the three beehives from different sites, or different farms, or different beekeeping systems (i.e. migratory and stationary)?

Why did authors mention the application of such monitoring device, if then the weight of the beehives is not used further? Indeed, within the climate index proposed, the bees’ activity is not derived using such data, rather using the number of days with a temperature higher than 10°C (as reported in line . As far as I understood reading the manuscript, the outputs of the wages’ measurements have not been used at all.

Concerning the statistical analysis

I have very little confidence in the statistical analysis performed. It is not clear which data have been statistically analysed (line 225). No explanation or justification is given concerning the type of statistical analysis used. Authors decided to apply a one-way ANOVA. Did they first verify the data distribution? This fundamental information is omitted.

Due to the very superficial description of the statistical analysis (which is even scattered in different sections, such as section 2.3 and section 3.2), I would suggest a deep general revision of the whole statistical analyses and a deep modification of the writing of this section.

Additionally, the results of the statistical correlation are reported in a table. However, they could be represented in a much clear manner, such as presenting graphs with trend lines.

3)      The verification of the results of this study is almost impossible since authors do not provide fundamental data such as the number of beehives investigated and the input-flows used to model the life cycle of the beekeeping systems, as well as about weather parameters of the beehives sites. Authors state that supplementary materials exist (mentioned in line 360) but I didn’t find them anywhere, nor at the end of the downloadable version of the manuscript, nor in any section of the review report form of the journal.

4)      I believe that an important restructuring of the paper would be beneficial, since the text suffer of lack of linearity, clarity and concepts are often popping up without any clear connection and explanation. For example, the majority of the study is focused on the CF (or LCA) performed, then suddenly the concepts of “wildflower honey climate index”, “effects of weather” and “precision agriculture principle” are mentioned disconnected one each other. This is reflected also in the abstract and in the introduction.

Furthermore, certain issues are reported under wrong sections (e.g. the development of the index, the “wildflower honey climate index” as authors named it, and the methodological description of the correlation performed, are reported within the result section, instead of the materials and methods section (see lines 356-383); authors refers to the fact that they investigated 14 apiary sites only in the results section (line 386).

5)      Last but not least, a deep revision of the English language should be performed since there are several poorly understandable and incorrect sentences (e.g. line 92: “The article builds upon previous research…, by performing climate data and LCA analysis…”; line 57 “Besides, beekeeping provides market products….”; line 281-282 “…are due to the shifting of input management in the two analysis years at farm….”; line 247-248 “We also correlation wildflower honey yield with ….”).

Furthermore, authors are invited to carefully read their work before submitting it. Beside the fact that the whole manuscript lack of clarity, it still contains sentences from the article template provided by the journal. For example, the description of what results should include provided by the journal is still present in the manuscript (see lines 251-253), as well as the example of what supplementary materials should contain (lines 554-555).

I came away with too many negative observations and questions to be able to recommend this paper for publication as it stands.

Therefore, I recommend authors to completely reformulate and redefine the aim of their work, diversifying it from the one already published by the same authors (i.e. Pignanoli et al. 2021), highlighting in clearer way what are the novelty elements present in their study compared with the knowledge currently present in the literature, and rewrite the manuscript, considering and undertaking the comments suggested.

Author Response

Author's Reply to the Review Report (Reviewer 2)

Authors of this manuscript attempts to assess the GHG emissions of beekeeping, distinguishing between two different beekeeping management (i.e. migratory and non-migratory systems). Furthermore, they develop an index linking some precipitations parameters and bees’ activity (number of days with a temperature higher than 10°C). Such index is correlated with the then wildflower honey yields and with the GHG emissions found.

To my opinion, these are topics of relevance and general interest to the readers of this journal. However, the manuscript in the current form is not acceptable for publication.

I list hereunder the main general issues that weaken the work.

1)      The novelty of this work is limited. Indeed, several life cycle assessment (LCA) analysis and carbon footprint (CF) of honey production have been published in literature (as appear in the references’ list) and the correlation between weather parameters (such as precipitation and temperature) with honey production has been also investigated by several scholars.

Authors - In this paper we aimed to quantify the GreenHouse Gas (GHG) emissions of honey production by performing the LCA methodology for the Climate Change (CC) Impact Category; to our knowledge there are still few studies that apply LCA methodology to beekeeping compared with other agri-food products. We considered an innovative aspect to assess the GHG emissions for honey production in three Italian regions affected by different climate conditions and also located at different altitudes; this beekeeping area has never been the target of LCA analysis.

In light of the influence of climatic factors on honey production and the important role of honey yield on the estimation of GHG emissions by LCA assessment, we have found it interesting to develop a climate index that crosses climate parameters with honey production.

Authors clearly state that the current work builds upon a previous work of them (Pignagnoli et al. 2021, cited by authors). Checking the paper of Pignagnoli et al. 2021 it appears that the analysis performed concerning the GHG emissions evaluation is almost the same, but including data from a second year (the ones of 2021). The results of 2020 of the present work are the same of the one published by Pignagnoli et al. 2021. Furthermore, the entire text of section 2.2.1 is almost identical of the one present in section 2.4 of Pignagnoli et al. 2021. I found too high similarity of several other sentences and paragraph between the two manuscripts.

The fact that a part of the results has been already published in another journal, as well as that entire sentences are almost the copy of the ones present in a published article, doesn’t represent the standards of publication ethics requested by this Journal.

Authors - The current work is based on the LCA methodology and it differs from the previous one mainly for data collection, data Inventory and database.

As regards the data collection, the differences are as follows:

  • Transport: taking into account the important role of this input in the results, for 2021 year the exact mileage is obtained from data really gathered from the machine odometer and reported in the beekeepers’ questionnaires. Based on this data, some trips for the 2020 year were re-calculated.
  • Supplementary feeding: the exact formula was re-calculated for two analysis years based on the dosage indicated on the product label, considering only the percentage of sugar on the product used.
  • Electricity consumption: we obtained this information directly from the meters in the laboratories of each farm and not from the bills, as it was done in the previous article. In fact, the cost of electricity in Italy has changed a lot in the last two years, so the conversion from euro to kWh would not always be exact and simple.

We have decided to change the data collection way in order to obtain a higher quality of data input, avoiding approximations by beekeepers. These differences led to different quantities of electricity, transportation, and supplementary food that are the main inputs in beekeeping supply chain. According to your good suggestion, more details about data collection were included in Section 2.2.

As regards the Life Cycle Inventory, the differences are as follows:

  • Waste that appears in the data inventory in the first article (Table 2) hasn’t been quantified in the current one (Supplementary Table 1). In fact in the farms analysed plastic and paper waste is recycled and therefore it doesn’t need to be accounted for LCA analysis.
  • Beekeeping equipment and machines, in this category we have not taken into account the impact of beekeeping equipment that according to beekeepers have a lifetime equal or even greater than 50 years.

As regards the Database, in the current work we used Agribalyse together with Ecoinvent, unlike the previous paper where we used only Ecoinvent 2.2 one. We found that the Agribalyse database v.3.0.1 was developed specifically for the agricultural sector and agrifood products whereas the EcoInvent 2.2 database is more suitable for the industry sector.

In addition, it should be noted that the objective of the two papers was also different; the data presented in Pignagnoli et al., 2021 were collected and elaborated for the GHG emissions quantification of the individual honey type, whereas in the current work it was assessed by mean honey production data. Based on the average beekeeping data, the climate index was developed.

The data are therefore original and the results of 2020 year in current work (Table 2, section 3.1) differ from previous work (Pignagnoli et al., 2021, Section 3.1). 

2)      I found serious inaccuracies and imprecisions about the methodology description, in both the GHG emissions evaluations and in the index definition, which make me doubt about the accuracy and correctness of their implementation.

Concerning GHG emissions evaluation

Carbon footprint (CF) and life cycle assessment (LCA) are different concepts and different methodology are used to evaluate them. In the whole document these terms are instead used interchangeably, confusing the reader and making unclear what the aim of the study is. Indeed, first it is said that the aim of the study is to conduct a LCA (e.g. line 22, 142-143), but later (e.g line 24, 145-156) carbon footprint is mentioned. Or even “In order to assess the environmental impact of honey production, ..” (line 153) making the reader struggling to understand to which environmental impact authors are referring to.

A definition of CF is given by authors in line 148-149 (although not in a clear form), but it is not what they assessed, since they didn’t include only “carbon dioxide emission” (line 148), rather certain GHG emissions (such as CO2, CH4 and N2O), as stated in line 153-155.

In the case authors performed a CF, they are invited to consult and follow the recent standard ISO 14067: 2018, which has been developed specifically concerning the Carbon footprint of products.

In the case authors performed a LCA, providing the results only for the climate change (or global warming) impact category, they are invited to clearly state it, using the right wording and definitions.

Authors - Suggestion accepted, we had better explain this concept in the test in the methodology section as well as in the abstract and the results ones. We performed an LCA analysis considering Climate Change Impact Category, which allows us to compare our results with other studies in the literature. The same approach is followed by Mujica et al., 2016. According to this change we have also modified the title.

In addition to this, at line 142-144 it is said that recent updates at the international level have been considered concerning the methodology used. To which updates are they referring to?

Authors - Suggestion accepted, we have included more details about the updates presented in the current paper compared to the previous one.

In LCA, as well as in CF, the functional unit is defined as “ performance of a product system for use as a reference unit. …. This functional unit defines what is being studied. All subsequent analyses are then relative to that functional unit, as all inputs and outputs in the LCI and consequently the LCIA profile are related to the functional unit.” (ISO 14044), that means “the functional unit that names and quantifies the qualitative and quantitative aspects of the function(s) along the questions “what”, “how much”, “how well”, and “for how long”. For a product this could be e.g. "Complete coverage of 1 m2 primed outdoor wall for 10 years at 99.9 % opacity").” (ILCD-Handbook 2010). Indeed, the FU used by authors is 1kg of honey in one year, and not kg CO2eq/kg honey, as wrongly reported by authors in line 154.

Authors - Suggestion accepted, sorry for this type of error. We have included a specific section (Section 2.3), in which we explained the FU as 1kg of honey.

Authors probably confuse and mixed up the inventory phase with the impact assessment phase. They named section 2.2.2 as Life Cycle Inventory Assessment (LCIA), which doesn’t correspond to any LCA phase. The ISO norms, which authors declare to follow, indicate four mains fundamental LCA phases: goal and scope definition, inventory analysis (or life cycle inventory - LCI), impact assessment (or life cycle impact assessment – LCIA) analysis and interpretation. Section 2.2.2 should be correctly named (i.e. life cycle impact assessment) and only information concerning the impact assessment phase should be provided (e.g., the fact that the database Agribalyse was used and that royal jelly and propolis was produced only by farm 1 (lines 215-223), should positioned in a section where the inventory is presented and discussed).

Authors - According to your good suggestions, we redrafted the Section 2 ’Materials and methods’; we have divided the chapters into ‘Goal and Scope definition’, ‘System Boundaries and Function Unit’, ‘ Life Cycle Inventory (LCI), ‘Allocation’ , ‘Life  Cycle Impact Assessment (LCIA)’. We have maintained the information about Agribalyse database, as Vásquez-Ibarra et al., 2021.

 Generally, LCA studies, as well as CF studies, report a quantitative list of input-flows (e.g. number of hives, quantity of wood used for each beehive, transport distances and frequency, fuel consumptions, electricity consumption, etc). None of this information is provided, making the analysis impossible to be replicated and verified. Also the whole beekeeping management systems are poorly described.

Authors - Thanks for the suggestion, we have included in Supplementary Tables 1, 2  and 3 the information required.

 Only one farm produces royal jelly, representing a peculiar situation among the farms investigated. To produce royal jelly, beehives are managed in a completely different way compared to ones dedicated to honey production. Why authors decided to include them in the assessment? I would suggest excluding the beehives dedicated to royal jelly production from the analysis, since the core object of the study is honey, and not royal jelly or the whole bee-farm products.

Authors - The farms concerned produced very small quantities of royal jelly and for this purpose, used relatively few hives. Especially Farm 1  produced in 2020 1 kg of royal jelly compared to total 3537 kg of products (honey + Beeswax + Honeycombs + Propolis + Royal jelly); in 2021 the same Farm 1 produced 0.40 kg of royal jelly compared to total 2621 kg of products (honey + beeswax + honeycombs + propolis + royal jelly).

Briefly, the hive model used for cells breeding is described as follows:  a central queen-less sector, composed by 6 honey-combs, separated, through queen-excluder grids, by two side queen-right sectors composed by 10 honey-combs.

The lateral compartments were also managed for the production of honey, placing on them the supers during the main nectar flows. According to this management, from the same hives Farm1 could achieve royal jelly as well as honey; So, the property allocation value was applied on them.

For these reasons we believe that the modest production of royal jelly has not significantly affected the management for the production of honey.

 Concerning allocation, since already in Pignagnoli et al. 2021 emerged that the honey yields significantly affect the results, authors could have estimated and presented the result using an additional FU, such as for example 1kg of beehives product in one year. In this way allocation would have been avoided, as suggested by the ISO and by several authors.

Authors - Thank you for your good suggestion. Unfortunately this kind of data is not currently in our possession and it is difficult now to get it from farms for past production years. According to the data obtained from beekeepers’ survey we have focused the assessment on honey production.

 Concerning the index development

The “wildflower honey climate index” is mentioned along the manuscript without giving any introduction about it. Indeed, within the introduction, as well as along the whole text, no basic and useful information are given, information on possible similar indexes are not provided to the reader, and an explanation on the usefulness of this index is missing.

Authors - Suggestion accepted, thank you. We specified in the text the validation made and the limits of the use of the developed index. We have inserted the usefulness of the development of the index and the bibliographic references have been useful (we have added them).

Several papers exist showing the influence of weather conditions (e.g. precipitation, temperature, relative humidity, wind speed, solar radiation) on honey production and on bees activities

It would be beneficial to develop further the index proposed in this study, including more weather-linked parameters and other fundamental aspects affecting honey production, such as the presence and severity of pathogens and disease, the availability and abundance of food sources (e.g. pollen, nectar, or honeydew produced by other insects), the ecological structure of the surrounding (e.g. fragmented landscape, natural areas, cultivated fields, urban centres, etc), possible polluting sources, etc. Some inspirations can be taken for example from additional recent literature:

Gounari, S., Proutsos, N., & Goras, G. (2022). How does weather impact on beehive productivity in a Mediterranean island?. Italian Journal of Agrometeorology 1: 65-81. https://doi.org/10.36253/ijam-1195;

Campbell, T.; Dixon, K.W.; Dods, K.; Fearns, P.; Handcock, R. (2020) Machine Learning Regression Model for Predicting Honey Harvests. Agriculture 10, 118. https://doi.org/10.3390/agriculture10040118;

and from Clarke, D., Robert, D. (2018) Predictive modelling of honey bee foraging activity using local weather conditions. Apidologie 49:386–396. https://doi.org/10.1007/s13592-018-0565-3.

Authors - We started with the idea of developing an index that considered many of the factors suggested, unfortunately with the statistical methodology used we were only able to insert some weather variables and only for wildflower honey. We will cherish the suggestion for our future studies, although we have seen that it will be very complicated even in relation to the many cofactors who have been listed.

Authors state that the index was developed to predict the production of honey (line 32). Did authors tested in some way the index they developed? Are the predicted results matching the measured results?

Authors state that the monitoring wages were placed under only three beehives (lines 244-245). Why only three and not maybe six beehives, one for each farm? Or even better, one for each site? How these three beehives have been selected? Where the three beehives from different sites, or different farms, or different beekeeping systems (i.e. migratory and stationary)?

Why did authors mention the application of such monitoring device, if then the weight of the beehives is not used further? Indeed, within the climate index proposed, the bees’ activity is not derived using such data, rather using the number of days with a temperature higher than 10°C (as reported in line. As far as I understood reading the manuscript, the outputs of the wages’ measurements have not been used at all.

Authors - Suggestion accepted, we have removed from the text of the article the section on devices. The remark is correct and the sentence has been rewritten, thank you. In the project we had funds only to install the devices in three beekeeping farms. The data from the devices are only used to relate to the resumption of activity of the bees.

Concerning the statistical analysis

I have very little confidence in the statistical analysis performed. It is not clear which data have been statistically analysed (line 225). No explanation or justification is given concerning the type of statistical analysis used. Authors decided to apply a one-way ANOVA. Did they first verify the data distribution? This fundamental information is omitted.

Due to the very superficial description of the statistical analysis (which is even scattered in different sections, such as section 2.3 and section 3.2), I would suggest a deep general revision of the whole statistical analyses and a deep modification of the writing of this section.

Additionally, the results of the statistical correlation are reported in a table. However, they could be represented in a much clear manner, such as presenting graphs with trend lines.

Authors - Suggestion accepted, thank you. We have reported all the statistical methodology in one section (2.3 Statistical analysis). Prior to analysis, the data population normality was verified, we added the sentence in the text. We have also modified table 3 with graphs that represent the correlation.

3)      The verification of the results of this study is almost impossible since authors do not provide fundamental data such as the number of beehives investigated and the input-flows used to model the life cycle of the beekeeping systems, as well as about weather parameters of the beehives sites. Authors state that supplementary materials exist (mentioned in line 360) but I didn’t find them anywhere, nor at the end of the downloadable version of the manuscript, nor in any section of the review report form of the journal.

Authors - Suggestion accepted, thank you. We have included in the Supplementary Table 1 data about Life Cycle Inventory (CI) inputs and outputs, LCI values are reported in Supplementary Tables 2 and 3 and weather parameters in Supplementary Table 5.

The tables (text and supplementary) were inserted in a single file. They have now been divided into 2 files. Thank you.

 4)      I believe that an important restructuring of the paper would be beneficial, since the text suffer of lack of linearity, clarity and concepts are often popping up without any clear connection and explanation. For example, the majority of the study is focused on the CF (or LCA) performed, then suddenly the concepts of “wildflower honey climate index”, “effects of weather” and “precision agriculture principle” are mentioned disconnected one each other. This is reflected also in the abstract and in the introduction.

Furthermore, certain issues are reported under wrong sections (e.g. the development of the index, the “wildflower honey climate index” as authors named it, and the methodological description of the correlation performed, are reported within the result section, instead of the materials and methods section (see lines 356-383); authors refers to the fact that they investigated 14 apiary sites only in the results section (line 386).

Authors - According to your good suggestions we have totally reconstructed the structure of the paper, remodelling the Section 2 and specifying at several points in the text that we performed an LCA analysis considering the impact category of Climate Change (CC). In addition we have better explained in the discussion Section that the application of precision agriculture is only a suggestion that can be implemented in further studies. In addition, the information about apiary sites is mentioned previously.

Suggestion accepted, thank you. We specified the apiary sites number in M&M section (2.4 Development of a prediction index based on meteorological data and correlation between environmental impact and honey production).

5)      Last but not least, a deep revision of the English language should be performed since there are several poorly understandable and incorrect sentences (e.g. line 92: “The article builds upon previous research…, by performing climate data and LCA analysis…”; line 57 “Besides, beekeeping provides market products….”; line 281-282 “…are due to the shifting of input management in the two analysis years at farm….”; line 247-248 “We also correlation wildflower honey yield with ….”).

Furthermore, authors are invited to carefully read their work before submitting it. Beside the fact that the whole manuscript lack of clarity, it still contains sentences from the article template provided by the journal. For example, the description of what results should include provided by the journal is still present in the manuscript (see lines 251-253), as well as the example of what supplementary materials should contain (lines 554-555).

I came away with too many negative observations and questions to be able to recommend this paper for publication as it stands.

Therefore, I recommend authors to completely reformulate and redefine the aim of their work, diversifying it from the one already published by the same authors (i.e. Pignanoli et al. 2021), highlighting in clearer way what are the novelty elements present in their study compared with the knowledge currently present in the literature, and rewrite the manuscript, considering and undertaking the comments suggested.

Authors state that the index was developed to predict the production of honey (line 32). Did authors tested in some way the index they developed? Are the predicted results matching the measured results?

Authors state that the monitoring wages were placed under only three beehives (lines 244-245). Why only three and not maybe six beehives, one for each farm? Or even better, one for each site? How these three beehives have been selected? Where the three beehives from different sites, or different farms, or different beekeeping systems (i.e. migratory and stationary)?

Why did authors mention the application of such monitoring device, if then the weight of the beehives is not used further? Indeed, within the climate index proposed, the bees’ activity is not derived using such data, rather using the number of days with a temperature higher than 10°C (as reported in line . As far as I understood reading the manuscript, the outputs of the wages’ measurements have not been used at all.

Authors - We thank you for the careful consideration and comments on the previous presentation of our manuscript entitled 'Greenhouse Gases (GHG) Emissions from honey production: two years of survey in Italian beekeeping companies' (has been revised according to your suggestion). We have also thoroughly reviewed the document in response to your comments.  The article was also reviewed by a native English speaker.

Reviewer 3 Report

The study presents an interesting research regarding the environmental impacts through the honey production in Italy. Two main systems (Mingratory and Non-Migratory approaches) with 3 farms under each system were researched and results were adquately presented. Some comments/suggestions are outlined below for the authors' consideration. 

1. Lines 160-161: So, should we say the LCA for this study covers "Honey Production" to "Farm Gate"? 

2. Lines 262-263: Is this the average of 6 farm productions? It would be ideal to also break it down into individual farm impacts by electricity, transport and supplemental feeding. 

3. Line 270: What parts of the honey production contribute towards the increase of CF from 2020 to 2021? 

4. Line 285: Does this mean a less transportation between apiaries in the migratory method?

5 . Lines 321-322: The definitions of honey productive phase (winter) and non-productive phase (spring-summer) are not consistent with the definitions provided in Lines 164-165. Please double check and update. 

6. Line 237: The proposed wildflower honey climate index needs to be further elaborated to rationalize the appropriateness of the index. Furthermore, an example based on a scenario with actual figures can be provided to provide a better explanation of the equation. 

7: Line 403: Is this an average of two years? It is not clear how this figure is derived. 

8: Lines 427-428: It would be useful to provide the distances traveled for the Migratory system.

9: Line 441: It is understood that the supplemental feeding entails a series of activities. It would be worthwhile to outline the activities here. 

10: Lines 461-463: Considering alternate locations could be one approach to minimise the transport required, however, different locations may also impact on the production of honey. Hence, this needs to be viewed holistically. 

11: Line 472: Through this study, has evidence been presented to demonstrate the reduction of CF through the precision beekeeping? 

Author Response

Author's Reply to the Review Report (Reviewer 3)

The study presents an interesting research regarding the environmental impacts through the honey production in Italy. Two main systems (Migratory and Non-Migratory approaches) with 3 farms under each system were researched and results were adequately presented. Some comments/suggestions are outlined below for the authors' consideration.

  1. Lines 160-161: So, should we say the LCA for this study covers "Honey Production" to "Farm Gate"?

Authors - Suggestion accepted, we have added details (just below line 175, we seemed the most suitable place), thank you.

  1. Lines 262-263: Is this the average of 6 farm productions? It would be ideal to also break it down into individual farm impacts by electricity, transport and supplemental feeding.

 Authors - Suggestion accepted, we specified in the text where this information is included.

  1. Line 270: What parts of the honey production contribute towards the increase of CF from 2020 to 2021?

Authors - Suggestion accepted, we specified in the text the contribution of supplemental feeding.

  1. Line 285: Does this mean less transportation between apiaries in the migratory method?

Authors - Thank you for the suggestion, we included in the text the reference to Supplementary Table 2.

5 . Lines 321-322: The definitions of honey productive phase (winter) and non-productive phase (spring-summer) are not consistent with the definitions provided in Lines 164-165. Please double check and update.

Authors - Sorry for the typing error, the sentence has been rewritten as suggested, thank you.

  1. Line 237: The proposed wildflower honey climate index needs to be further elaborated to rationalize the appropriateness of the index. Furthermore, an example based on a scenario with actual figures can be provided to provide a better explanation of the equation.

Authors - Suggestion accepted. Thank you so much for the review, a typo was found that made the text and Table 3. conflicting. Added ranges (Index/honey production) that we believe can make the proposed index more readable.

7: Line 403: Is this an average of two years? It is not clear how this figure is derived.

Authors - Thank you for your suggestion, I included the reference (Table 3) in the text.

8: Lines 427-428: It would be useful to provide the distances traveled for the Migratory system.

Authors - Suggestion accepted. Thank you so much for the review, we have included this type of information in Supplementary Tables 2.

9: Line 441: It is understood that the supplemental feeding entails a series of activities. It would be worthwhile to outline the activities here.

Authors - Suggestion accepted. We have added more details about supplemental feeding in the text. With the voice ‘Supplemental feeding’ we estimated only the amount of feeding provided. More quantities of food could impact other inputs (e.g. transport), but this aspect was not analysed in the present study.

10: Lines 461-463: Considering alternate locations could be one approach to minimize the transport required, however, different locations may also impact on the production of honey. Hence, this needs to be viewed holistically.

Authors - Thank you for your good suggestion. At first we introduced it as a suggestion for subsequent studies, not strictly the focus of the present study. However it represents a general conclusion, as you rightly suggested. The text has been revised.

11: Line 472: Through this study, has evidence been presented to demonstrate the reduction of CF through the precision beekeeping?

Authors - Thank you for your question/suggestion. No results have been analyzed on this aspect. The text has been revised.

Round 2

Reviewer 1 Report

The major changes suggested by me have been addressed. A few minor improvements;

In line 67, please give the full "Life Cycle Assessment" name and then (LCA) in parenthesis

In line 150, please change to "Functional Unit"

Reviewer 2 Report

I can state that the changes I required during my previouse reviews need for sure more time respect the two weeks used by authors to address my comments. I reviewed the revised version, and I noticed that several comments have not been addressed within the text (es: the index description is still under the results sections instead of the methodology section, quantitative data related to the inventory are still missing, information regarding the inventory are still included in the wrong section (i.e. within the impact assessment), allocation factors are not clear at all (e.g. they state that in Farm 4, all the impact are allocated to beewax (because they give a 100% of allocation factor). It's unclear why then they have impacts allocated to honey, that should have a 0%)), the introduction is missing any argumentation regarding the "indexes", which is the second main aim of the study.   I still convinced that this paper is not ready to be accepted, and it needs significant changes, as already stated and required in my previous revision report.